# Variability of Accommodative Microfluctuations in Myopic and Emmetropic Juveniles during Sustained near Work

**DOI:** 10.3390/ijerph19127066

**Published:** 2022-06-09

**Authors:** Hanyang Yu, Junwen Zeng, Zhouyue Li, Yin Hu, Dongmei Cui, Wenchen Zhao, Feng Zhao, Xiao Yang

**Affiliations:** State Key Laboratory of Ophthalmology, Zhongshan Ophthalmic Center, Sun Yat-sen University, Guangzhou 510275, China; yuhy23@mail2.sysu.edu.cn (H.Y.); zeng163net@vip.tom.com (J.Z.); lizhy65@mail2.sysu.edu.cn (Z.L.); eddy06980094@163.com (Y.H.); sarah72@126.com (D.C.); sswen080812@163.com (W.Z.)

**Keywords:** myopia, juvenile, sustained near work, asthenopia, digital screen

## Abstract

Near work has been considered to be a potential risk factor for the onset of myopia, but with inadequate evidence. Chinese adolescents use digital devices for near work, such as study and entertainment purposes, especially during the COVID-19 pandemic. In this study, we investigated the influence of prolonged periods of near work on accommodative response, accommodative microfluctuations (AMFs), and pupil diameter between juvenile subjects of myopia and emmetropia. Sixty juveniles (30 myopes and 30 emmetropes) were recruited for the study. Participants were instructed to play a video game on a tablet PC at a distance of 33.3 cm for 40 min. Accommodative response and pupil diameter were measured with an open-field infrared refractometer in High-speed mode. Parameters of the subjects were measured once every 10 min, and analyzed by one-way repeated measure ANOVA for variation tendency. There were no significant differences between emmetropia and myopia groups with respect to age and sex (*p* > 0.05). The low-frequency component (LFC) of myopia gradually increased with time, reached a peak at 30 min, and then declined (*p* = 0.043). The high-frequency component (HFC) of myopia also reached a peak at 30 min (*p* = 0.036). Nevertheless, there was no significant difference in the LFC (*p* = 0.171) or HFC (*p* = 0.278) of the emmetropia group at each time point. There was no significant difference in the mean and standard deviation of the accommodative response and pupil diameter both in emmetropic and myopic juveniles. Compared with juvenile emmetropes, myopes exhibit an unstable tendency in their accommodation system for prolonged near work at a certain time point. Accommodative microfluctuations may be a sensitive, objective indicator of fatigue under sustained near work in juvenile myopes.

## 1. Introduction

Myopia has become a major public health challenge worldwide in recent decades, especially in East and Southeast Asia [1,2]. Some projections suggest that, by 2050, nearly 50% of the global population may be myopic, and approximately 10% will be highly myopic [3]. Children with early-onset myopia face a greater risk of high myopia, which is often associated with sight-threatening pathologies [4]. Myopia is thought to be the result of the interaction between the environment and genetics. Associations between myopia and educational pressure are consistently observed, while associations between near work, increased screen time, and myopia are weak and inconsistent, although limitations on screen time are increasingly being considered as interventions to control the epidemic of myopia [5].

Near work has been considered to be a critical risk factor in the onset and development of myopia [6]; however, there is little direct evidence for the assertion. This uncertainty has instigated several studies on the accommodative response of myopic individuals.

Since hyperopic defocus induced by negative lenses or refractive surgery results in eye growth and myopia, as demonstrated in animal studies, it is suggested that accommodative lag during near work might generate the axial elongation of the eye in children [7,8,9,10]. However, several studies have demonstrated that no increased accommodative lag was present before myopia [11,12], and it appears to be a consequence rather than a cause of the development of incident myopia.

Nevertheless, other factors have been taken into consideration. When viewing a stationary target, the accommodation of the eye is not stable but constantly varies at a rapid speed within a range of approximately 0.50 D, a phenomenon termed: accommodative microfluctuations. Fluctuations at lower temporal frequencies (<0.6 Hz) are likely to be important for the control process of accommodation, which may make use of the associated changes in the spatial frequency spectrum and the contrast of the retinal image. The AMFs show a high degree of correlation of the two eyes, and the AMFs were similar under binocular and monocular observing conditions when the target conditions were the same [7,13,14]. Two dominant frequencies were revealed by power spectrum analysis of the microfluctuations waveform, including a low-frequency component (LFC) of <0.6 Hz and a high-frequency component (HFC) ranging between 1.0 and 2.3 Hz. The definition of HFC has appeared slightly inconsistently across different studies [15,16,17]. Interestingly, children with early-onset myopia and adults with late-onset myopia exhibit greater instability of accommodation compared with emmetropic subjects at the same stimulus levels [18,19], suggesting that accommodative microfluctuations may be a risk factor for the development of myopia. The mechanism may be described as follows: First, even if the lag of accommodation is small, large and unstable accommodative microfluctuations could accumulate hyperopic defocus signals, which induce myopia [20,21]. Second, adolescents with unstable microfluctuations generate vague images on the retina, which cause relative visual form deprivation, and this might participate in the development of incident myopia [22,23]. 

Digital screen time has been considered to be a modifiable environmental risk factor that may increase the risk of myopia. A massive number of children begin to use mobile devices even before the age of one, and most of them own a digital device by the age of four [24]. However, associations between myopia and screen time have not been consistently reported. Nevertheless, more recent studies appear to show related trends, especially those published in 2014 and later, that may be related to the increase in screen time in more recent years [25]. In 2018, China’s Ministry of Education issued a notice limiting the screen time for computers and mobile devices among children and adolescents. The World Health Organization (WHO) and the Ministry of Education of Taiwan also provide guidelines to restrict screen time in terms of digital devices to prevent myopia among preschool-aged children [26,27]. Since the underlying mechanism is still unclear, it is necessary to resolve this issue at the research level.

Previous studies focused on the role of microfluctuations in the accommodation system. However, only a few investigations have been carried out on accommodative behaviors during sustained periods of reading books or watching digital screens in natural visual environments. For a better understanding of the relationship between near work, accommodation, and the development of myopia, the current study aimed to investigate the differences in accommodative behaviors between myopic and emmetropic children during a prolonged period of digital near work, and to determine the optimal duration of digital near work in Chinese children.

## 2. Methods

### 2.1. Subjects

This clinical study was conducted at Zhongshan Ophthalmic Center (ZOC) in Guangzhou, China, after obtaining approval from the ZOC ethical committee prior to the initiation of the study. All experiments adhered to the Declaration of Helsinki. Written informed consent was obtained from the children and their parents or guardians. Sixty subjects aged between 7 to 12 years with a refractive error ranging from emmetropia to moderate myopia participated in the study, and all of them had a visual acuity of 0.1 logMAR or better with normal binocular vision. Children with astigmatism or anisometropia more than 1.0 D, those with amblyopia or manifesting strabismus, or those with a history of significant ocular disease and surgery were excluded from the study. Children who had undergone keratoplasty or were medicated with atropine in the past two years for controlling the progression of myopia were eliminated from the study. Emmetropia was defined as a mean spherical equivalent refractive (SER: sphere + (0.5 × cyl) between −0.50 and +0.50 D, and subjects with SER < −0.50 D were considered myopic.

### 2.2. Measurement Procedures

During the test, myopic subjects were fully corrected with suitable contact lenses. All of them had a visual acuity of 0.1 logMAR or better with the contact lenses. Participants were instructed to close their eyes and rest for at least 20 min to ensure relaxed accommodation and the stability of tear film after wearing their best-suited corrected visual distance contact lenses. Previous research has shown that there is no significant effect of contact lenses on the measurement of accommodation microfluctuations [28].

Subjects were asked to play a link game with a wireless mouse connected to a tablet computer (HUAWEI-M5) (1920 × 1200 pixels) at a distance of 33.3 cm corresponding to accommodative demands of 3.0 D. During the visual task, the luminance of the visual display terminal (brightness) was 210 cd/m^2^ (average) and the illumination of the background was 300 lux. Subjects were instructed to play the game in a usual manner. The visual display terminal was adjusted to be fixed on the machine in front of the participant at 33.3 cm for a conserved angular subtense within 5° horizontal and 5° vertical. Subjects were asked to concentrate on the game and maintain their head in the chin-rest throughout the 40 min of play.

Previous studies have confirmed that 3–5 min adaption is enough for accommodation [29,30,31]. Montani [32] also found that tear film was unchanged at 20 min wear of all lenses but was significantly reduced after 8 h. Hence, the 20 min rest is more than sufficient for accommodation and tear-film adaption. Sanders and McCormick pointed out that when reading a book or paper-like material, the normal reading distance is usually somewhere between 305 and 406 mm, with a mean of 355 mm. In the present study, children aged from 7 to 12 were instructed to play the game in a usual manner. The operation distance of tablet PC was usually closer than the typical desktop computer (nearly 400–500 mm) and is close to the distance of reading books [33,34]. In addition, the length of a child’s arm is shorter than those of an adult. The participant conducted the near task in 33.3 cm in a usual manner in order to match the confusion. We choose 300 lux for illumination of the background because it is also the hygienic standard for classroom lighting in primary and secondary schools in China (GB/T 7793-2010).

Only the left eye was measured while the task was performed in a natural, binocular state of vision. The accommodative response and pupil diameter were dynamically measured every 0.2 s (5 Hz) with an open-field instrument called Grand Seiko Auto Ref/Keratometer WAM-5500 (Grand Seiko, Hiroshima, Japan) in high-speed mode. The instruments have a sensitivity of 0.01 D and 0.1 mm for the accommodative response and pupil size, respectively [35,36]. Previous studies have confirmed the validity of the instrument’s ability to measure dynamical accommodation [29,35,37,38]. The power refractor was calibrated prior to the study.

The participants were encouraged to score in the game to remain focused. Pupil diameter and the accommodation response of the subjects were measured once at the beginning of the game, and taken as the baseline. Subsequently, the parameters of the subjects were measured once every 10 min. The participants were instructed to stop the game and stare at the midmost pattern in the screen with both eyes (the link game consisted of patterns of the same size during the process) to maintain the stability of the fixation. The subjects were asked to keep the pattern clear during the measurement and allow blinking. After a minute of continuous inspection, the participants continued to play the game. The measurements were repeated five times, and the data at the baseline, 10, 20, 30, and 40 min were obtained.

Microfluctuations vary with a series of factors, such as target luminance, spatial frequency, and stimulus demand; therefore, these factors should be taken into consideration while measuring the accommodative microfluctuations [37]. The above factors were controlled in the present study.

## 3. Data Analysis 

Data with no more than two blinks every 10 s were selected for analysis [39]. Blinks were automatically removed from the data by the HIGH-SPEED MODE software of the Grand Seiko WAM-5500. A one-dimension discrete Fourier was used to transform the data for power spectrum analysis. The sampling frequency is 5 Hz and the DFT points were 512, using a rectangular window. Our purpose was to ensure that the frequency domain signal post DFT is not distorted while achieving the required frequency resolution.

In the present study, the mean value and standard deviation of the accommodation response and pupil diameter, and microfluctuations of accommodation (including LFC and HFC) were compared by one-way repeated measure ANOVA within refractive groups. Only the data of the left eye were included in this study.

## 4. Results 

### 4.1. Baseline Data

The mean spherical equivalent of the myopia group was −2.51 ± 0.79 D, and that of the emmetropia group was −0.14 ± 0.32 D. The HFC of myopia was larger than that of the emmetropia (*p* < 0.00). The age of the two groups had no statistical difference. The baseline values of LFC were not significantly different between emmetropia and myopia groups. There was no significant difference in the mean and standard deviation of the accommodation response and pupil diameter between the two groups (Table 1).

### 4.2. Microfluctuations of Accommodation

#### 4.2.1. Low-Frequency Component of the Accommodative Microfluctuations

For the myopia group, the LFC of accommodative microfluctuations was significantly different between the baseline, and the periods of sustained near work of 10, 20, 30, and 40 min, as revealed by one-way repeated measures ANOVA (*p* = 0.043). The baseline value of LFC was lowest during the near work test. The LFC of the myopia group gradually increased with time and reached the peak at 30 min before a gradual decline. There was no significant difference in the LFC of the emmetropia group at each time point (*p* = 0.171) (Figure 1, Table 2).

#### 4.2.2. High-Frequency Component of the Accommodative Microfluctuations

For the myopia group, the HFC of accommodative microfluctuations was significantly different between the baseline and periods of sustained near work of 10, 20, 30, and 40 min, as revealed by one-way repeated measures ANOVA (*p* = 0.036). In general, the HFC of the myopia group gradually increased with time and reached a peak at 30 min. There was no significant difference in the HFC of the emmetropia group at each time point (*p* = 0.278) (Figure 2, Table 2).

### 4.3. Accommodative Response

For the myopia group, the mean and standard deviation of the accommodative response were not significantly different between the baseline and periods of sustained near work of 10, 20, 30, and 40 min, as revealed by one-way repeated measures ANOVA (*p* = 0.096) (*p* = 0.357). Similarly, for the emmetropia group, the mean and standard deviation value of the accommodative response were not significantly different between the various time points (*p* = 0.079) (*p* = 0.354).

### 4.4. Pupil Diameter

For the myopia group, the mean and standard deviation of pupil diameter were not statistically significant between the baseline and periods of sustained near work of 10, 20, 30, and 40 min, as revealed by one-way repeated measures ANOVA (*p* = 0.398) (*p* = 0.357). Likewise, for the emmetropia group, the mean and standard deviation values of pupil diameter were not statistically significant between the various time points (*p* = 0.069) (*p* = 0.354).

## 5. Discussion

Two main findings were identified in the current study. First, the myopia group displayed markedly different accommodative microfluctuations, especially in LFC, which increased during digital near work over time, but reached a peak at approximately 30 min and then decreased. However, the emmetropic juveniles showed no significant changes in accommodative microfluctuations throughout the near task. Second, the study found that both myopes and emmetropes exhibited a stable manifestation in the mean and standard deviation of the accommodation response and the pupil diameter in the period of near work.

On the basis of our findings, we propose three possible explanations: First, compared with the emmetropic eyes, myopic eyes exhibited an unstable tendency accompanied by the extension of the near work time. Myopia is inclined to fatigue, compared with emmetropia, in prolonged digital near work; however, it is difficult to isolate the causes or effects of this change in myopia. Second, the accommodative microfluctuations of myopic juveniles reached a peak and then declined during sustained near work, suggesting that there may be a time point of highest fatigue during prolonged near work for the accommodation system. Third, consistent with previous studies, the mean and standard deviation of the accommodation response and the pupil diameter of myopia remained constant over a period of prolonged near work [40,41]. Microfluctuations of accommodation in the present study showed a significant difference at different time points. We tentatively put forward that accommodative microfluctuations (LFC and HFC) could be a sensitive, objective indicator of fatigue in sustained near work.

Previous research focused on the discrepancy of accommodation in myopic and emmetropic eyes at a certain time point, but rarely concentrated on accommodative behaviors of sustained near work. Woodman et al. [42] found that after continuous viewing for approximately 30 min, the eye axial of myopia was significantly extended compared with emmetropia, which may be due to the accommodation. In the present study, the accommodative microfluctuations of the myopic adolescents gradually increased and reached a peak in 30 min, and then showed a downward trend in the process of prolonged near work. Buehren [43] found that myopes showed greater levels of some high-order ocular wavefronts than emmetropes at the baseline measurement. The differences between two groups became larger following 2 h of reading. In the present study, the HFC of the accommodative microfluctuations of myopia was larger than that of emmetropia at the baseline, and the tendency was also different, suggesting that myopic eyes exhibit greater instability during continuous near work. To the best of our knowledge, the present study is the first to confirm the difference in the accommodative microfluctuations of myopia and emmetropia in adolescents during sustained near work. After adding a tablet computer to the open field by modification, we created visual circumstances closer to the real environment, and the stability of accommodation-related parameters was guaranteed.

It should be noted that although near work and digital devices have long been considered to be risk factors of myopia, there is little direct evidence. Our study aimed to emphasize the latent role of sustained digital near work in the pathogenesis of myopia by providing objective evidence of accommodation microfluctuations which vary at different time points in prolonged digital near work. Our study indicated that there may be an optimal time point for adolescents to perform sustained digital near work, which could be monitored by the sensitive, objective measure of accommodative microfluctuations.

There are some limitations in the present study. First, the apparatus used to assess accommodation was an open-field instrument with a frequency of 5 Hz, which may be insufficient for measuring microfluctuations in accommodation. A more positive result may be obtained by an instrument with a frequency of more than 25 Hz because of the dense data and a smaller standard deviation [44]. Although the frequency was not high, it was adequate to reflect the difference in sustained digital near work between myopic and emmetropic eyes, verifying the preliminary hypothesis. Second, the duration of near work may have been too short for the emmetropic eyes to generate variation in accommodative microfluctuations. A longer duration of near work is required while assessing the microfluctuations of accommodation in emmetropia. Although there were no changes in the emmetropic eyes within the limited time, our study primarily demonstrated that the accommodation system of the emmetropic eyes in prolonged digital near work was stable within 40 min, while myopic eyes showed a risk of fatigue. 

In brief, there are still many clinically valuable investigations that warrant our attention. First, it is indispensable to conduct prospective longitudinal studies with a large sample size of children in the process of emmetropization to analyze whether the changes in the accommodative microfluctuations in myopia are simple or causal. Second, the time points of highest fatigue should be further investigated in juveniles with different rates of myopia progression in order to provide clinical evidence for the optimal duration for adolescents of sustained near work.

## 6. Conclusions

The present study revealed that, compared with stable emmetropes, myopes exhibit an unstable tendency in their accommodation system for prolonged near work at a certain time point. The accommodative microfluctuations could be a sensitive, objective indicator of fatigue in sustained near work.

## Figures and Tables

**Figure 1 ijerph-19-07066-f001:**
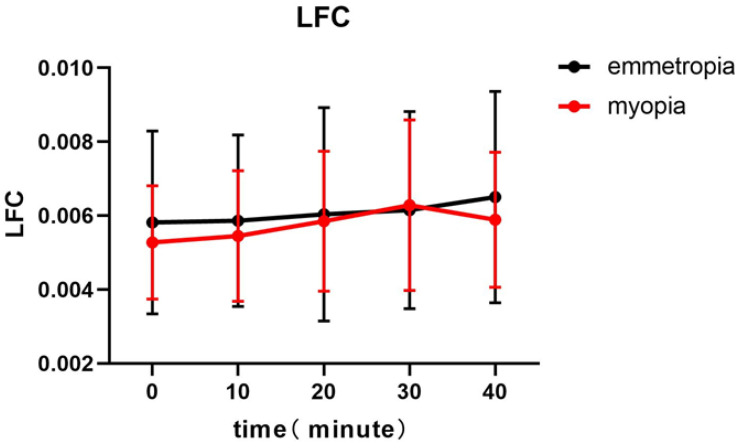
The baseline value of the low-frequency component (LFC) is lowest during the near work test. The LFC of myopia group gradually increases with time, reaching a peak at 30 min before a gradual decline. The low-frequency component of myopic individuals was significantly different between the baseline and the periods of sustained near work of 10, 20, 30, and 40 min, as revealed by one-way repeated measures ANOVA (*p* < 0.05).

**Figure 2 ijerph-19-07066-f002:**
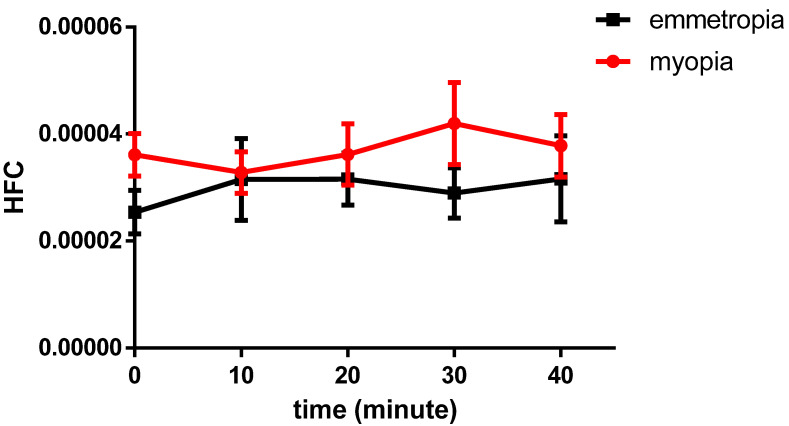
The HFC of the myopia group gradually increases with time and reaches a peak at 30 min, before a decline. The high-frequency component of myopic individuals was significantly different between the baseline and the periods of sustained near work of 10, 20, 30, and 40 min, as revealed by one-way repeated measures ANOVA (*p* < 0.05).

**Table 1 ijerph-19-07066-t001:** Descriptive statistics measured over the baseline duration of two refractive groups.

Baseline	Myopia		Emmetropia		*p*
	Mean	SD	Mean	SD	
Age (y)	10.96	2.83	10.23	2.17	0.266
SER (D)	−2.51	0.79	−0.14	0.32	**0.000**
LFC (Hz)	0.00528	0.00154	0.00582	0.00247	0.317
HFC (Hz)	0.0000361	0.0000106	0.0000254	0.0000109	**0.000**
Mean of AR (D)	−2.08	0.27	−2.01	0.61	0.537
SD of AR	0.30	0.12	0.24	0.15	0.106
Mean of PD (mm)	4.65	0.72	4.93	0.86	0.174
SD of PD (mm)	2.11	0.26	2.06	0.48	0.623

SER: spherical equivalent refraction; LFC: low-frequency component (0–0.6 Hz); HFC: high-frequency component (1.0–2.3 Hz); AR: accommodative response; PD: pupil diameter. Bold: Emphasize the significantly difference.

**Table 2 ijerph-19-07066-t002:** Descriptive statistics measured at baseline and after 10, 20, 30, and 40 min of sustained near work for two refractive groups.

	Group	10 Min		20 Min		30 Min		40 Min		*p*
		Mean	SD	Mean	SD	Mean	SD	Mean	SD	
LFC (Hz)	Myopia	0.00545	0.00177	0.00585	0.0019	0.00629	0.00231	0.00588	0.00183	**0.043**
	Emmetropia	0.00586	0.00233	0.00604	0.00289	0.00615	0.00267	0.0065	0.00286	0.171
HFC (Hz)	Myopia	0.0000328	0.0000104	0.0000362	0.0000153	0.000042	0.0000206	0.0000378	0.0000156	**0.036**
	Emmetropia	0.0000315	0.0000206	0.0000316	0.000013	0.000029	0.0000126	0.0000316	0.0000216	0.278
Mean of AR (D)	Myopia	−2.12	0.33	−2.22	0.31	−2.24	0.47	−2.21	0.31	0.096
	Emmetropia	−2.07	0.47	−2.08	0.59	−2.14	0.54	−2.18	0.53	0.079
SD of AR	Myopia	0.29	0.15	0.31	0.15	0.35	0.23	0.32	0.14	0.357
	Emmetropia	0.26	0.16	0.27	0.14	0.27	0.12	0.24	0.11	0.354
Mean of PD (mm)	Myopia	4.62	0.84	4.40	1.15	4.62	0.82	4.51	0.77	0.398
	Emmetropia	4.79	0.83	4.78	0.80	4.77	0.85	4.75	0.83	0.069
SD of PD (mm)	Myopia	2.15	0.30	2.25	0.30	2.30	0.39	2.24	0.29	0.357
	Emmetropia	2.10	0.45	2.11	0.56	2.16	0.52	2.19	0.53	0.354

LFC: low-frequency component (0–0.6 Hz); HFC: high-frequency component (1.0–2.3 Hz); AR: accommodative response; PD: pupil diameter. Bold: Emphasize the significantly difference.

## Data Availability

The data used to support the findings of this study are available from the corresponding author upon reasonable request.

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
