# Peer review of "Variability of Accommodative Microfluctuations in Myopic and Emmetropic Juveniles during Sustained near Work"

_ijerph, 2022, doi:10.3390/ijerph19127066_

Round 1
Reviewer 1 Report
The article Variability of Accommodative Microfluctuations in Myopic and Emmetropic Juveniles During Sustained Near Work is well structured, although there are some aspects that need to be improved and some corrections and/or observations that the authors should consider in order to improve the article.
I find the article interesting but it would be necessary to enrich the introduction. the state of the art should be enriched with references related to the difference in criteria on the causes of the myopia pandemic of the different researchers.
L61-65. Reference is made to decisions of Asian countries and the WHO on the limitation of children's time in front of screens , but no reference is made to the decision of those same countries ( Taiwan and China) to spend a mandatory number of hours on the air free every day. Perhaps the reception of more natural light, especially in the morning , may be more important for the decrease in the progression of myopia than the decrease in the use of screens.
L97-98. Explain why the distance of 33 cm is chosen for a Tablet. Why not choose 40-45 cm for example? Is it based on some bibliographical source, is it based on a test with the children prior to the experiment?
L98-99. The luminance of the visual display terminal (brightness) was 210 cd/m² (average) and the illumination of the background was 300 lux.
These light data are photometric. It would be interesting to know the amount of blue light that the screen emits and that of the interior lighting.
Would the results be the same if the light front is for example 2500 K as if it is 4000 K or 6500 K?
Is the accommodative effort in near vision the same if a small proportion of blue light enters the cornea (2500 K) than if a greater amount of blue light enters (6500 K)?
Is it the same to do this experiment in the morning with correct circadian balance after resting at night than to do it in the afternoon or at night?
Is an emmetropic person going to have the same accommodation and convergence process as a myopic person with lights with a greater amount of blue than with a lesser amount of blue?
I do not dispute that the results of this work are correct with the conditions that have been carried out, but my opinion is that not taking into account the spectral distribution of light and the intensity in each part of the spectrum means that the value of the results is probably much higher. less than expected.
I think it would be interesting to see if the results would be the same if they are done at different times of the day (24 hours) and with different illuminance conditions.
Author Response
RESPONSE TO REVIEWER 1
- I find the article interesting but it would be necessary to enrich the introduction. the state of the art should be enriched with references related to the difference in criteria on the causes of the myopia pandemic of the different researchers.
RESPONSE:
We are grateful for the suggestion. It is unfavorable for readers to understand the article while this part is missing. We would like to add the following content in the first paragraph of introduction.(seen in revised manuscript)
Myopia has become a major public health challenge worldwide in recent decades, especially in East and Southeast Asia[1, 2]. Some projections suggest that by 2050, nearly 50% of the global population may be myopia, and about 10% will be highly myopia[3]. Children with early onset myopia tend towards face a greater risk of high myopia which often associated with sight-threatening pathologies[4]. Myopia is thought to be the result of the interaction of environment and genetics. Associations between myopia and educational pressure are consistently observed in various researches while associations between near work, increased screen time and myopia is weak and inconsistent although limitations on screen time are increasingly under consideration as interventions to control the epidemic of myopia[5].
- L61-65. Reference is made to decisions of Asian countries and the WHO on the limitation of children's time in front of screens, but no reference is made to the decision of those same countries (Taiwan and China) to spend a mandatory number of hours on the air free every day. Perhaps the reception of more natural light, especially in the morning, may be more important for the decrease in the progression of myopia than the decrease in the use of screens.
RESPONSE:
Thanks for this comment.
Increased outdoor exposure time is a protective factor for myopia, independent of physical activity and near work[6]. Rose also reported that the risk of myopia is lower for adolescent who conduct a lot of near work while spend a lot of time outdoors than those who spend less time outdoors and work a lot at near in the meantime[7]. In 2010's tiantian120, Wu[8] added outdoor lighting during recess (a total of 120 minutes of outdoor activities per day) based on the previous myopia control strategy, the incidence of myopia has decreased by 1% annually since 2012-2015. It appears that time of outdoor rather than near work is the key factor for myopia control. However, it is still difficult to eliminate the possible damage to the onset and progression of myopia caused by near work (including prolonged digital near work). The purpose of this article is to present the specific changes of eye accommodation parameters in sustained near work. After all, accommodation is the major physiological change of eye when looking at near-sightedness, which is thought to be related to the occurrence of myopia[9]. And from a practical point of view, after clearly understand the optimal time point for sustained near work of adolescents, we can encourage young people to take outdoor activities during the recess (such as tiantian120), which may better ensure the control effect of myopia.
- L97-98. Explain why the distance of 33 cm is chosen for a Tablet. Why not choose 40-45 cm for example? Is it based on some bibliographical source, is it based on a test with the children prior to the experiment?
RESPONSE:
Thanks for this good question.
We choose 33 cm for the digital screen distance attribute to several reasons. ANSI/HFS (Human Factors Society) proposed that the VDT viewing distance should be greater than 300 mm in 1988. Sanders and McCormick also pointed out that when reading a book or paper-like material, the normal reading distance is usually somewhere between 305 and 406 mm, with a mean of 355 mm. In the present study, children age from 7 to 12 was instructed to play the game in a usual manner. Although less feelings of visual fatigue were associated with further VDT distance, the operation distance of tablet PC usually closer than the desktop computer(nearly 400-500mm) and is close to the distance of reading books[10, 11]. In addition, the length of the child’s arm is shorter than those of the adult. The aim of this study is to observe the variation during prolonged near work with time, and to compare the difference of myopia and emmetropia in accommodation behavior of sustain near work. The participate conduct the near task in the same distance in order to match the confusion.
- L98-99. The luminance of the visual display terminal (brightness) was 210 cd/m² (average) and the illumination of the background was 300 lux.
4.1 These light data are photometric. It would be interesting to know the amount of blue light that the screen emits and that of the interior lighting.
4.2 Would the results be the same if the light front is for example 2500 K as if it is 4000 K or 6500 K? Is the accommodative effort in near vision the same if a small proportion of blue light enters the cornea (2500 K) than if a greater amount of blue light enters (6500 K)? Is an emmetropic person going to have the same accommodation and convergence process as a myopic person with lights with a greater amount of blue than with a lesser amount of blue?
RESPONSE:
The reviewer’s statement is correct in that the variation of eye accommodation during a prolonged near work at different blue light level and color temperature is interesting. However, the author wish to determine the changes in eye parameters of adolescents while they executed a prolonged near work. Since Tatsiana and Redondo[12, 13] has demonstrated that the lag and variability of accommodation were insensitive to the blue light level, we concentrate on the discrepancy of accommodation behavior between myopia and emmetropia in a constant blue light level for 40 minutes.
4.3 Is it the same to do this experiment in the morning with correct circadian balance after resting at night than to do it in the afternoon or at night?
RESPONSE:
All of our examinations were conducted between 9.00-12.00am on weekends, which corresponds to the period when students carry out sustain near work in school on weekdays. Subjects were required to have a regular schedule and with adequate rest before the examinations. Thanks for your suggestion, it would be interesting to do a similar experiment at night. Unfortunately, subjects in this age often do not provide very good compliance at night, since they usually have to rest early to prepare for school or cram school in the next day. On the other hand, the main purpose of this study is to observe the variation during prolonged near work with time, and to compare the discrepancy of myopia and emmetropia in accommodation behavior of sustain near work. Therefore, we selected the weekend 9.00-12.00 time period for unified inspection, in order to match the confusion.
5.0 I do not dispute that the results of this work are correct with the conditions that have been carried out, but my opinion is that not taking into account the spectral distribution of light and the intensity in each part of the spectrum means that the value of the results is probably much higher. less than expected.
I think it would be interesting to see if the results would be the same if they are done at different times of the day (24 hours) and with different illuminance conditions.
RESPONSE:
The reviewer’s statement is correct in that the variation of eye parameters during a prolonged near work at different times of the day and with different illuminance conditions is interesting, the authors wish to divide our works into several stages. In clinical practice, parents often ask the doctor how long should the child take a break while execute a prolonged near work. Some researchers proposed a 10 minutes outbreak after 30 minutes sustained near work, but lack of direct evidence. The aim of this study is to determine the changes in eye parameters of adolescents while they executed a prolonged near work. On this basis we would change the illuminance and spectrum of the ambient light in order to find the interesting variation of eye parameters in a prolonged near work. The corresponding work would conduct in the near future and will publish it at a later time. The present study is the piror study of follow-up research.
Reference
- Dolgin E: The myopia boom. Nature 2015, 519(7543):276-278.
- Morgan IG, French AN, Ashby RS, Guo X, Ding X, He M, Rose KA: The epidemics of myopia: Aetiology and prevention. Progress in retinal and eye research 2018, 62:134-149.
- Holden BA, Fricke TR, Wilson DA, Jong M, Naidoo KS, Sankaridurg P, Wong TY, Naduvilath TJ, Resnikoff S: Global Prevalence of Myopia and High Myopia and Temporal Trends from 2000 through 2050. Ophthalmology 2016, 123(5):1036-1042.
- Haarman AEG, Enthoven CA, Tideman JWL, Tedja MS, Verhoeven VJM, Klaver CCW: The Complications of Myopia: A Review and Meta-Analysis. Investigative ophthalmology & visual science 2020, 61(4):49.
- Morgan IG, Wu PC, Ostrin LA, Tideman JWL, Yam JC, Lan W, Baraas RC, He X, Sankaridurg P, Saw SMet al: IMI Risk Factors for Myopia. Investigative ophthalmology & visual science 2021, 62(5):3.
- Wu PC, Tsai CL, Wu HL, Yang YH, Kuo HK: Outdoor activity during class recess reduces myopia onset and progression in school children. Ophthalmology 2013, 120(5):1080-1085.
- Rose KA, Morgan IG, Ip J, Kifley A, Huynh S, Smith W, Mitchell P: Outdoor activity reduces the prevalence of myopia in children. Ophthalmology 2008, 115(8):1279-1285.
- Wu PC, Chen CT, Chang LC, Niu YZ, Chen ML, Liao LL, Rose K, Morgan IG: Increased Time Outdoors Is Followed by Reversal of the Long-Term Trend to Reduced Visual Acuity in Taiwan Primary School Students. Ophthalmology 2020, 127(11):1462-1469.
- Logan NS, Radhakrishnan H, Cruickshank FE, Allen PM, Bandela PK, Davies LN, Hasebe S, Khanal S, Schmid KL, Vera-Diaz FAet al: IMI Accommodation and Binocular Vision in Myopia Development and Progression. Investigative ophthalmology & visual science 2021, 62(5):4.
- Jaschinski-Kruza W: Visual strain during VDU work: the effect of viewing distance and dark focus. Ergonomics 1988, 31(10):1449-1465.
- Jaschinski-Kruza W: Eyestrain in VDU users: viewing distance and the resting position of ocular muscles. Human factors 1991, 33(1):69-83.
- Palavets T, Rosenfield M: Blue-blocking Filters and Digital Eyestrain. Optom Vis Sci 2019, 96(1):48-54.
- Redondo B, Vera J, Ortega-Sanchez A, Molina R, Jimenez R: Effects of a blue-blocking screen filter on accommodative accuracy and visual discomfort. Ophthalmic & physiological optics : the journal of the British College of Ophthalmic Opticians (Optometrists) 2020, 40(6):790-800.

Reviewer 2 Report
- Clearer definition of accommodative microfluctuation is needed in Introduction.
- Suggest give justifications for the setting in measurement procedures. For example, explain why 20 min rest would ensure relaxed accommodation. Justify the choice of 33.3cm, 300 lux background, and setting of the display.
- What were the sampling rate and resolution of the keratometer?
- Give details of the setting of the DFT. For example, how many points? Was windowing used?
- Figure 1: Should vertical axis be labelled LFC instead of LFM?
- The frequency and magnitude of accommodation microfluctuation depend on many other factors: ANS balance, rate and depth of respiration, mean size of the pupil, and brightness of the environment. Did the presented methodology and analysis of results take these into consideration? For example, the game playing could have changed the ANS balance, which in turn affected the LFC and HFC.
- Due to the biomechanics structure of the pupil, the proportion of LFC and HFC would be affected by the mean pupil size. Was this taken into consideration?
- Suggest correct the grammatical errors and typos. Suggest improve formatting and technical writing style.
Author Response
RESPONSE TO REVIEWER 2
The co-authors and I would like to thank you for the time and effort spent in reviewing the manuscript. These comments are all valuable and very helpful for revising and improving our paper, as well as the important guiding significance to our research. We have studied comments carefully and have made corrections which we hope to meet with approval.
- Clearer definition of accommodative microfluctuation is needed in Introduction.
RESPONSE:
Thanks for yours comment on this problem. We intend to enrich the definition of accommodative microfluctuation in introduction Lines 49-57 as follow.(seen in revised manuscript)
When viewing a stationary target, the accommodation of the eye is not stable but constantly varies in a rapid speed within a range of approximately 0.50 D, which is termed accommodative microfluctuations. Fluctuations at lower temporal frequencies (<0.6 Hz) are probably important to the control process of accommodation, which may make use of the associated changes in the spatial frequency spectrum and contrast of the retinal image. The MFs show a high degree of correlation of the two eyes, and the MFs were similar under binocular and monocular observing conditions when the target conditions were the same.
- Suggest give justifications for the setting in measurement procedures. For example, explain why 20 min rest would ensure relaxed accommodation. Justify the choice of 33.3cm, 300 lux background, and setting of the display.
RESPONSE:
Thanks for your suggestion. We intend to enrich the justifications for the setting in measurement procedures as follow with your permission.
We instruct the participate to close their eyes and rest for at least 20 min before the experiment in order to ensure relaxed accommodation and stability of tear film. In fact, 3-5minutes adaption is enough for accommodation[1-3]. Montani[4] also found that tear film was unchanged at 20 minutes wear of all lenses but was significantly reduced after 8h. Hence, the 20 minutes rest is more than sufficient for accommodation and tear film adaption.
We choose 33 cm for the digital screen distance attribute to several reasons. ANSI/HFS (Human Factors Society) proposed that the VDT viewing distance should be greater than 300 mm in 1988. Sanders and McCormick also pointed out that when reading a book or paper-like material, the normal reading distance is usually somewhere between 305 and 406 mm, with a mean of 355 mm. In the present study, children age from 7 to 12 was instructed to play the game in a usual manner. Although less feelings of visual fatigue were associated with further VDT distance, the operation distance of tablet PC usually closer than the desktop computer(nearly 400-500mm) and is close to the distance of reading books[5, 6]. In addition, the length of the child’s arm is shorter than those of the adult. The aim of this study is to observe the variation during prolonged near work with time, and to compare the difference of myopia and emmetropia in accommodation behavior of sustain near work. The participate conduct the near task in the same distance in order to match the confusion.
We choose 300 lux for illumination of the background. Because it is as well as the hygienic standard for classroom lighting in primary and secondary schools in China(GB/T 7793-2010). we aim to have a better grasp of the accommodation behavior of adolescent in a usual manner.
- What were the sampling rate and resolution of the keratometer?
RESPONSE:
The sampling frequencies of the instruments used in this study has been shown in Lines 120-121 in the text. Accommodative response and pupil diameter were dynamically measured every 0.2 s (5 Hz). The instruments with a sensitivity of 0.01 D and 0.1 mm for the accommodative response and pupil size, respectively[7, 8].
- Give details of the setting of the DFT. For example, how many points? Was windowing used?
RESPONSE:
Sampling frequency: 5Hz
DFT Points: 512
Windowing: Rectangular window
Our purpose is to ensure that the frequency domain signal post DFT is not distorted while achieving the required frequency resolution.
- Figure 1: Should vertical axis be labelled LFC instead of LFM?
RESPONSE:
We are very sorry that our figure will confuse readers, and thanks again for your correction. We have modified the figure as your advice.
- Due to the biomechanics structure of the pupil, the proportion of LFC and HFC would be affected by the mean pupil size. Was this taken into consideration?
RESPONSE:
Yes, as you said, pupil size is one of the main factors affecting LFC and HFC. We have taken into consideration of this element before the experiment. As shown in table 1 and lines 151-152, there was no significant difference in the mean and standard deviation of the pupil diameter between the two groups at the baseline. (Table 1). And the mean and standard deviation values of pupil diameter were not statistically significant between the time point for both groups, as shown in table 2 and lines 199-203. We also found that were no statistically significant between the two groups in each time point, using the independent-samples T test (P>0.05).
- The frequency and magnitude of accommodation microfluctuation depend on many other factors: ANS balance, rate and depth of respiration, mean size of the pupil, and brightness of the environment. Did the presented methodology and analysis of results take these into consideration? For example, the game playing could have changed the ANS balance, which in turn affected the LFC and HFC.
RESPONSE:
As shown in lines 134-136 in the text. Microfluctuations vary with a series of factors which should be taken into consideration while measuring the accommodative microfluctuations. The above factors were controlled in the present study. To our best knowledge, there were no significant difference in ANS balance or rate and depth of respiration between the myopia and emmetropia teenager. In addition, recent and past work has found that, accommodation and other aspects of eye movements are connected. Pupil diameter varied along with the lens accommodation [9]. And the size of pupil was associated with the ANS balance, as demonstrated in previous research[10]. In the present study, the pupil diameter and accommodative response remained constant in both groups during the near work, implying that the ANS balance was relatively stable. All of the participate conduct a same near digital game in the same environment (such as: the same brightness of the environment which remained constant during the test). The physical condition was similar between the two groups of adolescents apart from the refractive state. But we did found the discrepancy of the accommodative microfluctuations between the two groups.
- Suggest correct the grammatical errors and typos. Suggest improve formatting and technical writing style.
RESPONSE:
Thanks for your suggestion, we will improve the grammar and writing skills of any sentences in the text that need to be revised. We are going to use editing service to help us to correct them.
Reference
- Day M, Gray LS, Seidel D, Strang NC: The relationship between object spatial profile and accommodation microfluctuations in emmetropes and myopes. J Vis 2009, 9(10):5 1-13.
- Seidel D, Gray LS, Heron G: Retinotopic accommodation responses in myopia. Investigative ophthalmology & visual science 2003, 44(3):1035-1041.
- Khan KA, Dawson K, Mankowska A, Cufflin MP, Mallen EA: The time course of blur adaptation in emmetropes and myopes. Ophthalmic & physiological optics : the journal of the British College of Ophthalmic Opticians (Optometrists) 2013, 33(3):305-310.
- Montani G, Martino M: Tear Film Characteristics During Wear of Daily Disposable Contact Lenses. Clin Ophthalmol 2020, 14:1521-1531.
- Jaschinski-Kruza W: Visual strain during VDU work: the effect of viewing distance and dark focus. Ergonomics 1988, 31(10):1449-1465.
- Jaschinski-Kruza W: Eyestrain in VDU users: viewing distance and the resting position of ocular muscles. Human factors 1991, 33(1):69-83.
- Sheppard AL, Davies LN: Clinical evaluation of the Grand Seiko Auto Ref/Keratometer WAM-5500. Ophthalmic & physiological optics : the journal of the British College of Ophthalmic Opticians (Optometrists) 2010, 30(2):143-151.
- Green W, Ciuffreda KJ, Thiagarajan P, Szymanowicz D, Ludlam DP, Kapoor N: Accommodation in mild traumatic brain injury. J Rehabil Res Dev 2010, 47(3):183-199.
- Dominguez-Vicent A, Monsalvez-Romin D, Del Aguila-Carrasco AJ, Ferrer-Blasco T, Montes-Mico R: Changes in the anterior chamber during accommodation assessed with a Scheimpflug system. J Cataract Refract Surg 2014, 40(11):1790-1797.
- Wu F, Zhao Y, Zhang H: Ocular Autonomic Nervous System: An Update from Anatomy to Physiological Functions. Vision (Basel) 2022, 6(1).

Reviewer 3 Report
In this study the authors have studied how various accommodation parameters change in myopes versus the emmetropes with sustained near work. They find significant differences within the low and high frequency components over time in the myopia group, which could potentially affect the progression of the disease in the myopia group.
While the authors have done a detailed analysis of the accommodation parameters in the two groups, I have one following comment-
Lines 68-70: “However, only a few investigations have been carried out…..” This is not true, there are several articles studying the effect of increased screen times in children in development as well as progression of myopia. Some of these are- Ip et al. (IOVS, 2008), Buehren et al. (Vision Research, 2005), Ku et al. (Ophthalmology, 2018), Saxena et al. (pLos One, 2015), Terasaki et al. (BMC Ophthalmology, 2017), Zhou et al. (2016) etc. The authors should include these references and also perhaps discuss how their results compare to these previously published reports in the Discussion.
Author Response
RESPONSE TO REVIEWER 3
The co-authors and I would like to thank you for the time and effort spent in reviewing the manuscript. These comments are all valuable and very helpful for revising and improving our paper, as well as the important guiding significance to our research. We have studied comments carefully and have made corrections which we hope to meet with approval.
- Lines 68-70: “However, only a few investigations have been carried out…..” This is not true, there are several articles studying the effect of increased screen times in children in development as well as progression of myopia. Some of these are-Ip et al. (IOVS, 2008), Buehren et al. (Vision Research, 2005), Ku et al. (Ophthalmology, 2018), Saxena et al. (pLos One, 2015), Terasaki et al. (BMC Ophthalmology, 2017), Zhou et al. (2016) etc. The authors should include these references and also perhaps discuss how their results compare to these previously published reports in the Discussion
RESPONSE:
The authors would like to thank the reviewer for the suggestion. Our statement may not be clear enough. There are many studies exploring the relationship between near work or electronic product use and myopia. However, some of these studies used questionnaires to explore the subject, which not only has the potential interference of recall offset on the results, but more importantly, we have no way of knowing the specific changes of the subjects' eye parameters during a continuous near work process.
Terasaki[1] found the correlation between incidence of myopia and the duration of electronic product use. However, Ku[2] found that cram school attendance, rather than the digital usage (Computer/Internet/video game) may increase the risk of children’s incident myopia. Notably, time spent in cram school (2.78 h/d) is much higher than those spent on reading (0.63 h/d) and use of computer/Internet and games (0.68 h/d) in this study. Therefore, it cannot be said that the use of electronic products will not promote the occurrence and progress of myopia. It is worth noting that since the epidemic of COVID-19, India, China and other countries have repeatedly closed the schools and used electronic products for online courses at home. In fact, electronic products may not pose a greater risk of myopia than traditional near work (ex: reading books), but may play a role in promoting myopia as traditional near work. However, unfortunately, heavy use of electronics for study, work and entertainment is gradually becoming the normalization.
We intend to enrich the discussion in Lines 232-237 as follow with your permission.
Buehren[3] has found that the myopes showed greater levels of some high order ocular wavefronts than the emmetropes at the baseline measurements. The differences between two groups became larger following 2 h of reading. In the present study, the HFC of accommodative microfluctuations of myopia was larger than the emmetropia at the baseline and the tendency was different as well, which implying that myopic eyes exhibit more instability properties during continuous near work.
Reference
- Terasaki H, Yamashita T, Yoshihara N, Kii Y, Sakamoto T: Association of lifestyle and body structure to ocular axial length in Japanese elementary school children. BMC Ophthalmol 2017, 17(1):123.
- Ku PW, Steptoe A, Lai YJ, Hu HY, Chu D, Yen YF, Liao Y, Chen LJ: The Associations between Near Visual Activity and Incident Myopia in Children: A Nationwide 4-Year Follow-up Study. Ophthalmology 2019, 126(2):214-220.
- Buehren T, Collins MJ, Carney LG: Near work induced wavefront aberrations in myopia. Vision Res 2005, 45(10):1297-1312.

Round 2
Reviewer 2 Report
In response to the reviewers' comments, the authors have provided further explanation and justifications on the cover page. However, most of these new contents (on the cover page) were not incorporated in the revised manuscript.
Author Response
2nd RESPONSE TO REVIEWER 2
In response to the reviewers' comments, the authors have provided further explanation and justifications on the cover page. However, most of these new contents (on the cover page) were not incorporated in the revised manuscript.
The co-authors and I would like to thank you for the suggestion. We would like to enrich the methods refer to the following items with your permission(seen in revised manuscript)
- Suggest give justifications for the setting in measurement procedures. For example, explain why 20 min rest would ensure relaxed accommodation. Justify the choice of 33.3cm, 300 lux background, and setting of the display.
L126-137. Previous studies have confirmed that 3-5minutes adaption is enough for accommodation[1-3]. Montani[4] also found that tear film was unchanged at 20 minutes wear of all lenses but was significantly reduced after 8h. Hence, the 20 minutes rest is more than sufficient for accommodation and tear film adaption. Sanders and McCormick pointed out that when reading a book or paper-like material, the normal reading distance is usually somewhere between 305 and 406 mm, with a mean of 355 mm. In the present study, children age from 7 to 12 was instructed to play the game in a usual manner. The operation distance of tablet PC usually closer than the desktop computer(nearly 400-500mm) and is close to the distance of reading books[5, 6]. In addition, the length of the child’s arm is shorter than those of the adult. The participate conducted the near task in 33.3cm in a usual manner in order to match the confusion. We choose 300 lux for illumination of the background because it is as well as the hygienic standard for classroom lighting in primary and secondary schools in China(GB/T 7793-2010).
- What were the sampling rate and resolution of the keratometer?
L141-142. The instruments with a sensitivity of 0.01 D and 0.1 mm for the accommodative response and pupil size, respectively[7, 8]
- Give details of the setting of the DFT. For example, how many points? Was windowing used?
L162-164.The sampling frequency is 5Hz and the DFT points were 512 using rectangular window. Our purpose is to ensure that the frequency domain signal post DFT is not distorted while achieving the required frequency resolution.

Round 3
Reviewer 2 Report
The reviewers' comments and suggestions for changes have been considered. There are still some minor problems with grammar and technical writing style.
Author Response
We had get a professional grammar review from recommended website.
Attachment is last English-Editing-Certification.
Thank you!
This manuscript is a resubmission of an earlier submission. The following is a list of the peer review reports and author responses from that submission.